# What to Do with the New Antibiotics?

**DOI:** 10.3390/antibiotics12040654

**Published:** 2023-03-27

**Authors:** Khalil Chaïbi, Françoise Jaureguy, Hermann Do Rego, Pablo Ruiz, Céline Mory, Najoua El Helali, Sara Mrabet, Assaf Mizrahi, Jean-Ralph Zahar, Benoît Pilmis

**Affiliations:** 1Département de Réanimation Médico-Chirurgicale, AP-HP Hôpital Avicenne, Université Sorbonne Paris Nord, F-93000 Bobigny, France; 2Common and Rare Kidney Diseases, Sorbonne Université, INSERM, UMR-S 1155, F-75020 Paris, France; 3Service de Microbiologie Clinique, AP-HP, Groupe Hospitalier Paris Seine Saint-Denis, F-93000 Bobigny, France; 4Equipe Mobile de Microbiologie Clinique, Groupe Hospitalier Paris Saint-Joseph, F-75014 Paris, France; 5Laboratoire de Microbiologie Clinique Et Plateforme de Dosage Des Anti-Infectieux, Groupe Hospitalier Paris Saint Joseph, F-75014 Paris, France; 6Plateforme de Dosage Des Anti-Infectieux, Groupe Hospitalier Paris Saint-Joseph, F-75014 Paris, France; 7AP-HP Hôpital Avicenne, Prévention du Risque Infectieux, GH Paris Seine Saint-Denis, F-93000 Bobigny, France; 8Institut Micalis UMR 1319, Université Paris-Saclay, INRAE, AgroParisTech, F-78350 Jouy-en-Josas, France

**Keywords:** multidrug-resistant bacteria, ceftolozane/tazobactam, ceftazidim/avibactam, imipenem/relebactam, meropenem/vaborbactam, cefiderocol

## Abstract

Multidrug-resistant Gram-negative bacteria-related infections have become a real public health problem and have exposed the risk of a therapeutic impasse. In recent years, many new antibiotics have been introduced to enrich the therapeutic armamentarium. Among these new molecules, some are mainly of interest for the treatment of the multidrug-resistant infections associated with *Pseudomonas aeruginosa* (ceftolozane/tazobactam and imipenem/relebactam); others are for carbapenem-resistant infections associated with *Enterobacterales* (ceftazidime/avibactam, meropenem/vaborbactam); and finally, there are others that are effective on the majority of multidrug-resistant Gram-negative bacilli (cefiderocol). Most international guidelines recommend these new antibiotics in the treatment of microbiologically documented infections. However, given the significant morbidity and mortality of these infections, particularly in the case of inadequate therapy, it is important to consider the place of these antibiotics in probabilistic treatment. Knowledge of the risk factors for multidrug-resistant Gram-negative bacilli (local ecology, prior colonization, failure of prior antibiotic therapy, and source of infection) seems necessary in order to optimize antibiotic prescriptions. In this review, we will assess these different antibiotics according to the epidemiological data.

## 1. Epidemiology of Resistance

The global rise of resistant bacteria threatens the effectiveness of the antibiotics that have transformed medicine and saved countless lives. In particular, multidrug-resistant (MDR) Gram-negative bacterial infections are a major global public health problem [1], leading to high morbidity and mortality [2,3]. In recent decades, the problem of antimicrobial resistance, particularly in Gram-negative bacteria, has challenged the management of serious nosocomial infections [4,5]. In 2017, to address the threat of antimicrobial resistance (AMR), the World Health Organization (WHO) developed and prioritized a list of antibiotic-resistant pathogens to guide research into and development of new and effective antibiotic treatments. A study evaluating the antibiotic susceptibility of over 20,000 clinical isolates was recently published [4]. Among the 3472 included *Pseudomonas aeruginosa* strains, 587 (16.9%) were considered MDR and 355 (10.2%) were extensively drug-resistant (XDR). Furthermore, among the 16,606 included *Enterobacterales* strains, 1269 (7.6%) were extended-spectrum beta-lactamase (ESBL) producers and 103 (0.62%) were resistant to carbapenems. In this study, the authors observed higher rates of resistance to certain antimicrobial agents in intensive care unit (ICU) patients. Higher resistance among *P. aeruginosa* and some *Enterobacterales* species was identified primarily against third-generation cephalosporins (3GC), carbapenems, and piperacillin-tazobactam.

Carbapenemases, the most versatile family of β-lactamases, occupy a prominent position in this problematic situation. *Klebsiella pneumoniae* carbapenemases (KPC), members of the Ambler class A (serine β-lactamase) classification, are the most prevalent and are found primarily on plasmids in *Klebsiella pneumoniae*. Class D carbapenemases consist of β-lactamases of the OXA subtype that are frequently identified in *Acinetobacter baumannii*. Metallo β-lactamases (IMP, VIM, GIM for instance) have been detected mainly in *P. aeruginosa*, but there are increasing reports of this group in the *Enterobacterales.*

New antibiotics with some activity against the main XDR pathogens are available, but their place in the therapeutic armamentarium remains to be defined. In this review, we will assess these different antibiotics according to the epidemiological data.

## 2. Microbiological Interest of These New Antibiotics

### 2.1. Ceftolozane/Tazobactam (CTZ)

Ceftolozane/tazobactam (CTZ) combines a novel cephalosporin with an established β-lactamase inhibitor. Ceftolozane is an oxyimino-aminothiazolyl cephalosporin with a pyrazole substituent at the 3-position side chain, which provides protection against hydrolysis by AmpC β-lactamases [5]. Ceftolozane/tazobactam is very effective against *Escherichia coli* (minimum inhibitory concentration (MIC)_50/90_, 0.25/0.5 mg/L; 98.5–99.9% inhibition at an MIC of 8 mg/L) and retains activity against some ESBL-producing strains (MIC_50/90_, 0.5/2–32 mg/L) [6]. It has activity against *Streptococcus* spp., limited activity against *Staphylococcus* spp., and no activity against *Enterococcus* spp. It is active against anaerobic bacteria but has limited activity against *Bacteroides* spp. *Acinetobacter* spp. and *Stenotrophomonas* spp. are usually resistant [7]. A study by Sader et al., including 10,532 Gram-negative bacilli (GNB) (2191 *P. aeruginosa*, of which 31.9% were MDR, 24.6% were XDR, and 11.6% were isolates susceptible only to colistin, and 8,341 *Enterobacterales*), observed that CTZ was the most active β-lactam molecule against *P. aeruginosa* (CTZ/TZ (MIC_50_, 1 mg/L) inhibiting 90% of isolates with a MIC of ≤8 mg/L) and showed better in vitro activity than the other cephalosporins (except cefiderocol) and piperacillin-tazobactam against MDR *Enterobacterales*, but superiority over meropenem and tigecycline was not demonstrated [8].

### 2.2. Ceftazidime/Avibactam (CZA)

Ceftazidime/avibactam (CZA) is a combination of ceftazidime and a new β-lactam inhibitor. Avibactam belongs to the diazabicyclooctane family, a structural class of inhibitors that does not contain a β-lactam core but retains the ability to covalently acylate its β-lactamase targets. This covalent linkage occurs by reversible acylation. In the case of other β-lactamase inhibitors, the reaction is irreversible and produces intermediates that are hydrolyzed. The novel β-lactamase inhibitor avibactam is a potent inhibitor of class A, class C, and some class D enzymes and decreases ceftazidime MIC; unfortunately, it has little activity against class B β-lactamases [9,10].

CZA has excellent activity against ESBL-producing *Enterobacterales* and AmpC-overproducing *Enterobacterales,* including strains that produce both enzymes. An average decrease from 64 µg/mL to 0.5 µg/mL has been shown in 34,062 *Enterobacterales* strains producing ESBL or overproducing AmpC [11]. It has activity against strains such as OXA-24 and OXA-48 [12], which are carbapenem hydrolyzing class D β-lactamases, and against CTX-M- and KPC-producing variants [13,14]. CZA is also active against *P. aeruginosa*, although one study showed an 18% level of resistant strains (resistance related to loss of porin and efflux pumps) [15]. CZA has limited activity against anaerobic bacteria [16].

### 2.3. Imipenem/Relebactam (IMI/REL)

Carbapenems are known for their activity against β-lactamases, including class A ESBLs and class C β-lactamases (AmpCs). Relebactam is a potent non-β-lactam, bicyclic diazabicyclooctane, β-lactamase inhibitor. The addition of relebactam maintains or enhances imipenem activity against Gram-negative bacteria [17]. Zhanel et al. compared imipenem alone versus imipenem and relebactam. Relebactam was found to significantly increase activity against imipenem-non-susceptible isolates and β-lactamase-producing (class A enzymes) *Enterobacterales* and against *P. aeruginosa* (2- to 128-fold MIC reductions for *Enterobacterales* and an 8-fold MIC reduction for *P. aeruginosa*) [17]. Karlowsky et al. compared imipenem and IMI/REL against 12,170 *P. aeruginosa* strains and found that the combination of relebactam decreased the MIC of imipenem (MIC_50/90_; 0.5/2 for the combination and 2/>32 for imipenem alone) [18]. Unfortunately, it had little activity against *Acinetobacter baumannii*, *Chryseobacterium*, and *Stenotrophomonas maltophilia* [17].

In summary, IMI/REL is an effective antibiotic against class A and KPC-producing *Enterobacterales*, AmpC-producing *P. aeruginosa*, and OprD loss *P. aeruginosa* (by inducing a downregulation of the OprD expression) [17]. It has no activity against metallo-β-lactamase, such as VIM-, IMP-, and NDM-producing bacteria [17].

### 2.4. Meropenem/Vaborbactam (MEV)

Meropenem has a spectrum of activity close to that of imipenem, except that it has better activity against *P. aeruginosa* and lower activity against *Enterococcus faecalis*. Vaborbactam is a non-β-lactam, cyclic boronic acid with a high affinity for serine β-lactamases [17]. A recent study by Castanheira et al. collected 14,304 Gram-negative isolates. Against carbapenem-resistant *Enterobacterales* (n = 265) MDR strains (n = 1210) and XDR strains(n = 161), MEV showed MIC_50/90_ values of 0.5/32, 0.03/1, and 0.5/32 g/mL, whereas the meropenem activities were 16/32, 0.06/32, and 0.5/32 g/mL, respectively [19]. Furthermore, the addition of vaborbactam had a limited impact on *Acinetobacter baumannii* and *P. aeruginosa.* It improves the activity of meropenem against class A and class C β-lactamases, and vaborbactam did not potentiate the activity of meropenem against *Enterobacterales* producing OXA-48-like β-lactamases. In general, the isolates with high MEV MICs contain multiple resistance mechanisms, such as class B and D β-lactamase, which are not inhibited by vaborbactam, porin alterations, or efflux pump overexpression.

### 2.5. Cefiderocol (CFD)

Cefiderocol is a synthetic conjugate composed of a cephalosporin part and a catechol-type siderophore that uses a “trojan-horse approach” to bind to iron and facilitate bacterial cell entry using active iron transporters. In the periplasmic space, it dissociates the iron part and binds PBP 3 to inhibit bacterial cell synthesis. This mechanism allows for increased efficiency against efflux pumps or porin loss resistance mechanisms [20]. Cefiderocol is a broad spectrum antibiotic with activity against *Enterobacterales* and nonfermenting bacteria, such as *Pseudomonas* spp., *Acinetobacter* spp., *Burkholderia* spp., and *Stenotrophomonas maltophilia* [21]. Shortridge et al. tested CFD against GNB collected from hospitalized patients in the United States and Europe in 2020, as part of the SENTRY Antimicrobial Surveillance Program. The GNB isolates included 8,047 *Enterobacterales*, 2282 *P. aeruginosa*, 650 *Acinetobacter* species, and 338 *Stenotrophomonas maltophilia*. They found that the MIC_90_ of cefiderocol was lower for carbapenem-resistant *Enterobacterales*, MEV, or IMI/REL-resistant *Enterobacterales* [22]. Cefiderocol also has in vitro activity against various bacteria, such as *Haemophilus* spp., *Moraxella* catarrhalis, and *Bordetella parapertussis*, but shows relative activity against *Campylobacter jejuni* and ceftriaxone-resistant *Neisseria gonorrhoeae* [21].

## 3. What Is to Be Learnt from the Published Studies?

### 3.1. Ceftolozane/Tazobactam (CTZ)

This new antibiotic was marketed in France in 2016 on the basis of three randomized controlled trials evaluating its efficacy in clinical practice for the treatment of pyelonephritis and complicated urinary tract infections, ASPECT-cUTI [23]; complicated intra-abdominal infections, ASPECT-cIAI [24]; and nosocomial pneumonia, ASPECT-NP [25].

These studies included *Enterobacterales* strains with variable proportions of ESBL and *P. aeruginosa*, some of which were MDR strains. It was particularly effective against *P. aeruginosa*, with a 100% clinical cure in the ASPECT-cIAI study (11 of which had overproducing AmpC, and 13 were considered MDR) [26]. In a companion study of ASPECT-NP (5), CTZ prevented the emergence of resistance in *P. aeruginosa*, whereas 25.9% of the strains in the meropenem group became resistant [27].

In retrospective studies that included MDR- or XDR *P. aeruginosa*-related infections, the CTZ arm a showed better clinical cure or reduced mortality compared to several antibiotic regimens, including Colistin [28,29,30].

A study comparing CTZ with Meropenem in ESBL or derepressed cephalosporinase-producing *Enterobacterales* is ongoing (MERINO-3 study) [31].

### 3.2. Ceftazidime/Avibactam (CZA)

This new molecule was evaluated in four pivotal studies (REPRISE [32], RECAPTURE [33], RECLAIM 3 [34], and REPROVE [35]). These studies included urinary tract infections, pneumonia, and complicated intra-abdominal infections, some of which included MDR/XDR *Enterobacterales* (ESBL-producing, AmpC-overexpressing, or carbapenemase-producing strains). In these studies, the clinical cure rate was higher in the CZA arm for the carbapenemase-producing strains [36,37].

A post hoc analysis of these four trials that analyzed MDR strains (including *Pseudomonas*) found a higher cure rate in the CZA arm than in the comparator arm [38].

For the carbapenemase-producing *Enterobacterales*-related infections (including KPC- and OXA-48-producing strains), several retrospective studies have compared CZA with colistin [39,40,41,42] or tygecycline [43], showing a better clinical cure rate with CZA.

More recently the CAVICOR trial [44], which is a retrospective multicenter study including 349 patients treated for carbapenemase-resistant *Enterobacterales*-related infection, compared CZA with the best available treatment (amikacin, tobramycin, tigecycline, fosfomycin, and colistin in mono- or combination therapy) and showed lower 30-day mortality and higher clinical or microbiological cure in the CZA treatment arm.

### 3.3. Imipenem/Relebactam (IMI/REL)

This novel combination was evaluated in two randomized controlled trials. The RESTORE-IMI [45] study demonstrated the non-inferiority of IMI/REL compared to imipenem and colistin for the treatment of imipenem-resistant GNB-related infections. The RESTORE-IMI-2 study demonstrated non-inferiority IMI/REL compared to piperacillin-tazobactam in the treatment of nosocomial pneumonia with respect to 28-day mortality in [46].

### 3.4. Meropenem/Vaborbactam (MEV)

Two pivotal trials have evaluated MEV. The TANGO I study [47] demonstrated the non-inferiority or even superiority of MEV compared to piperacillin/tazobactam for a clinical and microbiological cure. The TANGO II study [48] showed a better clinical and microbiological cure without differences in mortality in the treatment of carbapenem-producing *Enterobacterales* compared to the best available therapy (including CZA, monotherapy, or a combination with tetracyclines, carbapenem, colistin, or aminoglycosides).

A retrospective study showed similar efficacy between MEV and CZA for the treatment of KPC-producing *Enterobacterales*-related infections [49]. Meropenem/vaborbactam may be effective against KPC-producing *Enterobacterales* resistant to CZA [50,51,52].

### 3.5. Cefiderocol (CFD)

In the APEKs-cUTI study, CFD was shown to be superior to imipenem in terms of a clinical and microbiological cure for the treatment of urinary tract infections [53] and to be non-inferior to imipenem in the treatment of nosocomial pneumonia (APEKS NP) [54] in terms of 14-day mortality.

CREDIBLE-CR is the most recent open-label, randomized, controlled trial involving carbapenem-resistant GNB. Cefiderocol was compared with the best available antibiotic therapy (mono- or combination therapy including colistin) [55].

On several endpoints, according to the site of infection (a clinical or microbiological cure), the results were similar between the two arms of treatment. However, there was an excess mortality in the CFD group (34% vs. 18% at the end of the study) in patients whose source of infection was not identified. Finally, two retrospective studies showed the superior efficacy of CFD compared to colistin for the treatment of carbapenem-resistant *Acinetobacter* sp. [56,57].

A multicenter non-inferiority study evaluating CFD for the treatment of healthcare-associated and hospital-acquired bloodstream infections is currently in progress [58].

### 3.6. Aztreonam + Ceftazidim/Avibactam

There are few therapeutic options for the treatment of metallo-β-lactamase-producing *Enterobacterales*-related infections. A recent meta-analysis [59] confirmed the efficacy of the innovative association between aztreonam and CZA.

In an open-label prospective study [60] that included NDM- and VIM-producing *Enterobacterales*-related bloodstream infections, a combination of aztreonam and CZA reduced 30-day mortality compared with the best available therapy (including colistin, tigecycline, aminoglycosides, and fosfomycin).

## 4. For Which Patient to Prescribe These New Antibiotics as Empirical Treatment?

### 4.1. “It Takes All the Running You Can Do, to Keep in the Same Place”

This quote from Lewis Carroll’s *Through the Looking-Glass* summarizes the history of antibiotics from the accidental discovery of penicillin in the late 1920s to the development of new antibiotics. The evolving response to antibiotic resistance has often been compared to the famous red queen’s race described in the novel [61]. This situation, combined with the growing concern beyond the medical community regarding the consequences of anti-microbial resistance (recognized as a threat to human health as great as climate change [62]), has led to increased awareness of the stewardship of antibiotic therapy. While this awareness is commendable, the drivers of this “race” may not be as simple as they seem. Recent European data have shown that, with the exception of some Eastern European countries, carbapenem resistance is unlikely to be solely due to its consumption [63] and that other factors (strain impact and microbiota state for instance) should be considered.

In the global context of the willingness to save carbapenems, these elements should be considered when considering the rationale for using these new antibiotics. On the other hand, it may be incongruous to discuss the probabilistic indication of recently marketed antibiotics. The marketing authorization could not be considered sufficient when most of these drugs were granted in the context of an emergency. However, the purpose is not to venture recommendations but to discuss the literature data on this topic and to provide insights for the future.

### 4.2. Rationale for an Empirical Use of Antibiotics

Empirical therapy should consider the type of microorganism, the site of infection, the patient’s history and condition, the local epidemiologic data, and the pharmacokinetic properties of the antibiotics [64]. Although some of these variables can be documented, there is still much uncertainty regarding the correct choice of antibiotics. Thus, empirical therapy requires a certain broadness of spectrum, especially with regard to critically ill patients. It may seem contradictory to discuss the probabilistic relevance of antibiotics that have proven to be effective only in specific cases. For instance, although CZA and CTZ have excellent activity against some resistant bacteria, they have limited activity against anaerobic bacteria, which should be considered in most types of infections. CTZ has insufficient activity against KPC. Although the most international recommendations recommend limiting the use of new molecules to microbiologically documented infections, their use in empirical treatment should be discussed [65,66]. In fact, the use of these molecules can be discussed as probabilistic therapy before the identification of antibiotic-resistant bacteria. These situations include critically ill patients in the ICU setting, according to the local prevalence of antibiotic resistance among *P. aeruginosa* or *Enterobacterales*; patients who fail therapy with a carbapenem; specific patient populations known to be at high risk for infection with MDR bacteria (e.g., those immunocompromised due to a solid organ transplant); and patients with previous infection or colonization with MDR *P. aeruginosa* or *Enterobacterales* (Figure 1). Furthermore, numerous studies suggest that a preliminary colonization step is a mandatory prerequisite for the development of infection related to MDR Enterobacterales, but the transition from colonization to infection remains rare [67]. Currently, knowledge of carrier status complicates the choice of empirical antibiotic therapy, especially in patients without clinical severity criteria, because it exposes the patient to either the risk of over-consumption of broad-spectrum antibiotics or the risk of antibiotic therapy inadequacy.

### 4.3. What Is the Hurry?

Decades of debate have been triggered by the difficult balance between the need to choose the appropriate antibiotic and the need to delay treatment. Although many aspects should be considered, the patient’s condition seems to be the major factor influencing the risk of mortality in cases of delayed initiation of antibiotic therapy [68]. Therefore, the timing of antibiotic initiation (and its probabilistic nature) should be focused on the population of patients with severe conditions.

In this specific population, studies have shown that the earlier CZA [44] and CTZ [44,69] are initiated, the more survivors there are. Some studies have also shown that the last resort use of these antibiotics could lead to a loss of opportunity [44,69]. Nevertheless, these studies have led to the documentation of bacteria and antibiotic susceptibility. As stated earlier, it is already known that treating severe patients as soon as possible with the correct treatment is an effective strategy. However, it does not address the issue of treatment without bacterial identification or documentation of antibiotic susceptibility test results. Only a study comparing carbapenem with new antibiotic introduction in critically ill patients with suspected (but undocumented) ESBL or *P. aeruginosa* would answer this question. To our knowledge, there are no data to suggest that one of these antibiotics is an effective carbapenem-sparing strategy in a probabilistic strategy

### 4.4. In Clinical Practice (Table 1)

Specific and rare cases of carbapenem failure in a probabilistic context with a strong suspicion of infection with a germ sensitive to these new antibiotics could be discussed on a case-by-case basis. However, the WHO classified most of these new antibiotics in the “reserve” category, which stands for “antibiotics that should only be used as a last resort when all other antibiotics have failed” and most of the guidelines strongly advise against an empirical use of these new antibiotics in this indication [44,63]. Further clinical trials are required to assess the relevance of their empirical use. Finally, the rapid availability of antibiotic susceptibility tests has been shown in several studies to reduce the duration of empirical treatment and the duration of therapeutic inadequacy [44,69].

**Table 1 antibiotics-12-00654-t001:** Preferred and alternative treatment recommendations (adapted from IDSA 2022 [66] and ESCMID 2022 guidelines [65]) for carbapenem-resistant *Enterobacterales*.

	*Enterobacterales*			Multidrug *Pseudomonas aeruginosa*
	Classe A KPC	Classe B NDM, VIM	Classe D Oxa-48	
Ceftolozane/tazobactam				1st intention
Ceftazidime/avibactam	1st intention		1st intention	Possible
Imipeneme/relebactam	Possible			Possible
Meropeneme/vaborbactam	Possible			
Aztreonam/avibactam	Possible	1st intention	Possible	Possible
Cefiderocol	Possible	Possible	Possible	Possible

Red: ineffective.

## 5. Disadvantage of These New Antibiotics?

### 5.1. Emergence of Resistance 

Even though novel β-lactam/β-lactamase inhibitor combinations (βL-βLIC) are last-resort antibiotic (ATB) options for the treatment of MDR Gram-negative infections, increasing reports of βL-βLIC resistance have been published in recent years. Resistance is due to several mechanisms, including enzyme mutations that alter the hydrolytic properties of β-lactamase, alteration of the antibiotic target or expression of an alternative target, changes in cell permeability (i.e., porin deficiencies), overexpression of efflux pumps, and/or increased carbapenemase expression. Few reviews have summarized these emerging reports of βL-βLIC resistance based on survey studies, individual case reports, or laboratory selection experiments [70,71,72]. 

#### 5.1.1. Ceftolozane/Tazobactam (CTZ)

*P. aeruginosa* and *Enterobacterales* isolates carrying class A (e.g., KPC) and class B carbapenemases (e.g., VIM, NDM, and IMP) are usually resistant to CTZ [73]. Several mechanisms of CTZ resistance have been described, including overexpression and mutations in class C β-lactamase (AmpC). In a review, Papp Wallace et al. summarized the various resistance mechanisms that have been reported [72]. *P. aeruginosa* produces a chromosomally encoded class C cephalosporinase (*Pseudomonas*-derived cephalosporinase [PDC]) that is often responsible for the resistance to β lactam antibiotics, but usually, PDCs are not efficient at hydrolyzing ceftolozane. Resistance to CTZ in class C β-lactamases other than PDC (e.g., CMY and FOX) has been reported much less frequently. The acquisition of several different class A β-lactamases (e.g., GES, PER-1, and VEB-1) has been associated with the emergence of CTZ resistance. OXA variants (eg, OXA-14), mainly due to a single amino acid substitution, have also been identified in *P. aeruginosa* isolates. Additional resistance mechanisms, such as decreased expression of oprD and/or up-regulation of efflux pumps, may contribute to the emergence of CTZ resistance in *P. aeruginosa* [72,74].

#### 5.1.2. Ceftazidime/Avibactam (CZA) 

CZA resistance has emerged in the USA and Europe and has been considered a serious cause for concern [75]. Several studies have reported the emergence of KPC mutations following antimicrobial therapy [76,77]. Outside of the CZA combination, it is inactive against class B (e.g., NDM, VIM, and IMP) β-lactamases due to the absence of an active-site serine residue and against Acinetobacter OXA-type carbapenemase (OXA-24/40) [73]. Mainly, resistance to CZA in *Enterobacterales* is caused by three different mechanisms: enzymatic alteration, modification of the antibiotic target, or changes in cell permeability (or expression of efflux pumps). Specific mutations within class A carbapenemases are the most common and well-characterized mechanisms associated with CZA resistance and have been frequently reported in KPC3-producing K. pneumoniae [71]. Furthermore, although resistance to CZA was due to mutations in the bla_KPC_ gene, the increased gene expression and copy number of mutated bla_KPC_ genes contributed to the highest MIC for CZA [78]. Resistance to CZA has also been reported in other class A (CTX-M, SHV, GES, and VEB), class C (Amp C), and class D (OXA) β-lactamase mutations [71,72]. In combination with antibiotic target modification, decreased expression and/or mutations in porin genes (e.g., OmpK35 and OmpK36), or alterations in efflux pumps, the CZA MICs increase significantly, ranging from 128 to 256 mg/L (e.g., a 5- to 7-fold increase in the initial CZA MICs) [79]. In *P. aeruginosa*, increased AmpC expression associated with the overexpression of the MexAB-OprM system [80] or with OprD loss [81] contributes to an increased CZA MIC.

#### 5.1.3. Imipenem/Relebactam (IMI/REL) 

*Enterobacterales* and *P. aeruginosa* that carry class B metallo-carbapenemases (e.g., VIM, NDM, and IMP) or class D OXA β-lactamases that are not susceptible to inhibition by imipenem or relebactam are also resistant to MEV [73]. Imipenem/relebactam resistance can also be due to various mechanisms, including carbapenemase mutation, carbapenemase overexpression, penicillin binding protein (PBP) mutation or underexpression, increased acrB efflux and decreased permeability (decreased expression and/or mutations in porin genes). Gaibani et al. reviewed the studies demonstrating this combination of resistance mechanisms among *Enterobacterales* [76]. 

Resistance to IMI/REL was mainly reported in *Enterobacterales* due to the loss of OmpK35 and OmpK36 and the hyperexpression of bla_KPC_. In contrast, in carbapenemase-negative *P. aeruginosa* isolates, reduced susceptibility to IMI/REL was mainly caused by OprD porin downregulation, whereas AmpC expression did not seem to affect IMI/REL MICs [82]. For *P. aeruginosa* producing ESBL enzymes (VEB, PER, GES, and SHV), a moderate reduction in activity was observed. In addition, isolates harboring GES-5 carbapenemase exhibited high IMI/REL MICs (ranging from 32 to 128 mg/L) [83]. 

#### 5.1.4. Meropenem/Vaborbactam (MEV) 

*Enterobacterales* and *P. aeruginosa* that carry class B metallo-carbapenemases (e.g., VIM, NDM, and IMP) or class D OXA β-lactamases and are not susceptible to inhibition by meropenem or tazobactam are also resistant to MEV [73]. The main mechanism of MEV resistance among *Enterobacterales* is impaired permeability due to porin loss/mutations associated with β-lactamase overexpression and increased efflux pump production [76]. Vaborbactam crosses the outer membrane using the OmpK35 and OmpK36 porins [81]. The OmpK36 porin, which has a smaller channel than OmpK35, appears to play a more important role in the influx of vaborbactam across the outer membrane [82]. Lapuebla et al. found that vaborbactam activity was reduced in KPC-Kp isolates with reduced expression of OmpK36 compared to the same KPC-producing isolates with functional porins. In addition, efflux pump systems, particularly AcrAB-TolC, are common resistance mechanisms against multiple antibiotic classes. Lomovskaya et al. showed that the downregulation of ompK35 and overexpression of acrAB (due to mutation in the ramR gene) did not affect the activity of MEV, whereas the overexpression of acrAB in association with inactivated ompK35 and ompK36 porins increased the MIC of MEV [84]. The effect of a combination of multiple resistance mechanisms against MEV in *K. pneumoniae* isolates was illustrated in a study conducted by Zhou et al. in 2018; the study showed that the MIC of MEV was not affected by reduced OmpK35 or increased expression of bla_KPC_ or acrB alone, whereas the strains with complete inactivation of porins in combination with increased expression of bla_KPC_ and acrB genes were associated with the highest MIC for MEV [85]. 

#### 5.1.5. Cefiderocol 

Cefiderocol appears promising as it can overcome most of the mechanisms of β-lactam resistance. Nevertheless, CFD resistance has been reported and is high in some cohorts (up to 50%) [70]. Karakonstantis et al. reviewed the mechanisms of resistance to CFD, the prevalence of heteroresistance, and the reports of emerging resistance in vivo [70]. Several mechanisms of resistance have been identified, including selected β-lactamases (mainly NDM1, KPC and AmpC variants, OXA-427, and PER- and SHV-type ESBLs), mutations affecting siderophore receptors, mutations affecting porins and efflux pumps, and mutations in PBP-3 (the target of cefiderocol). Each of these mechanisms alone is usually not sufficient to raise the CFD MIC above the pharmacokinetic/pharmacodynamic (PK/PD) breakpoints; this is further supported by the fact that various studies of clinical isolates have not correlated CFD resistance with specific mechanisms but have identified multiple resistance mechanisms predominantly involving co-expression of different β-lactamases in combination with permeability defects. The clinical impact of heteroresistance is that resistant subpopulations may emerge during treatment, leading to treatment failure and the spread of resistant strains. The high prevalence of heteroresistance to CFD has been proposed as an explanation for the suboptimal efficacy of CFD against carbapenem-resistant bacteria, particularly *A. baumannii* [86,87]. However, an increasing number of cases of the in vivo emergence of resistance during treatment have been reported [70,88,89,90].

### 5.2. Consequences on Microbiota, Collateral Effects 

Knowledge of the ecological consequences of new antibiotics is essential for the positioning of their indications in the therapeutic armamentarium. With the exception of a single study [91], there are no clinical or fundamental data available to classify various new molecules according to their ecological effects. The aim of this study was to investigate the effect of CZA on the human gut microbiota. Fecal samples were collected from 12 healthy volunteers receiving 6 g/24 h on days 1 to 6. Ceftazidime and avibactam concentrations in plasma (ceftazidime 0–224.2 mg/L of plasma and avibactam 0–70.5 mg/L of plasma) and feces (ceftazidime 0–468.2 mg/kg of feces and avibactam 0–146.0 mg/kg of feces) were determined by bioassay. A sample culture showed a significant decrease in *E. coli* and other *Enterobacterales* during CZA administration, whereas the number of enterococci increased. *Lactobacilli*, *Bifidobacteria*, *Clostridia*, and *Bacteroides* decreased significantly after CZA administration. *Lactobacilli* (*p* < 0.0001), *Bifidobacteria* (*p* = 0.0004), *Clostridia* (*p* = 0.0001), and *Bacteroides* (*p* = 0.0003) decreased significantly during CZA administration. The effects on *Lactobacilli*, *Bifidobacteria*, and *Bacteroides* were similar among the 12 volunteers, whereas *Clostridia* showed different ecological patterns among the volunteers. Toxigenic *Clostridioides difficile* strains were detected in five subjects during the study. Loose stools were reported as an adverse event in four cases. Another study evaluating the ecological consequences of a combination including avibactam [91] highlighted a decrease in *E. coli* and other *Enterobacterales* and *Enterococci* during treatment and an increase in *Klebsiella* sp. and *Candida*. All the bacterial counts were normalized between day 9 and day 21. Finally, the digestive diffusion of the inhibitors can make us fear an effect added to that of the antibiotic with which it is associated. 

## 6. How to Administer These New Antibiotics (PK/PD Criteria)

All of these antibiotics showed time-dependent activity. Thus, as with the other β-lactam antimicrobials, the PK/PD target is to maintain a free drug concentration at least above the MIC or 4–5 times above the MIC of the targeted pathogen for 100% of the dosing interval (100%fT > MIC or 100%fT > 4–5 × MIC), to avoid the selection of resistant mutants and to ensure antibiotic penetration into tissues and fluids at the infection sites [92,93].

The standard doses were 2 g/0.5 g tid in a 2 h infusion for CZA; 2 g/2 g tid in a 3 h infusion for meropenem/vaborbactam; 2 g/1 g tid in a 1 h infusion for CTZ; 500 mg/250 mg qid in a 30 min infusion for IMI/REL; and 2 g × 3/24 h in a 3 h infusion for CFD. Dose adjustment is recommended for patients with impaired renal function or increased renal clearance.

All these antibiotics exhibit a short elimination half-life of 1–3 h and a low protein binding ranging from 2% to 30% (except for cefiderocol, 58%). Their clearance is mainly renal, and more than half is excreted unchanged in the urine [94,95,96,97,98,99].

Studies in healthy subjects with standard doses have reported the rates and the extent of the plasma to lung distribution, described by the epithelial lining fluid/plasma concentration ratio, ranging from 0.3 (CZA) to 0.5 (IMI/REL) and 0.63 (MEV) [100,101,102].

In critically ill patients, pharmacokinetic exposure can be difficult to predict, and standard dosing regimens often fail to achieve optimal PK/PD targets. Therefore, more frequent injection or prolonged or continuous infusion (CI) should be preferred to optimize antibiotics dosing. The continuous infusion of CTZ and CZA with a loading dose has been successfully used for deep-seated infections, achieving high clinical and microbiological cure rates. Therapeutic drug monitoring (TDM) confirmed that CI achieved PK/PD targets throughout the dosing interval [103,104]. Meropenem/vaborbactam and CFD showed physicochemical stability in syringes in normal saline for 8h and 24h, respectively, opening the possibility of prolonged or continuous infusion [105].

In addition, to avoid the risk of therapeutic failure, probabilistic treatment should consider the clinical breakpoints of the pathogens according to EUCAST: CZA and MEV ≤ 8 mg/L; IMI/REL and CFD ≤ 2 mg/L for both *Enterobacterales* and *P. aeruginosa*; and CTZ ≤ 2 mg/L for *Enterobacterales* and ≤ 4 mg/L for *P. aeruginosa* [106].

TDM can guide dose optimization to ensure PK/PD target attainment and to minimize the toxicity of these antibiotics.

## 7. Conclusions

Since the early 2000s, the pipeline of novel antibiotics has dried up due to an unsustainable economic model, characterized by an insufficient return on investment to fund the research and development of new antibiotics. This dwindling antimicrobial research and development pipeline has been addressed by the WHO, the US Centers for Disease Control and Prevention (CDC), and many other stakeholders with funding solutions to develop novel therapeutics to treat MDR bacterial and fungal pathogens. In this context, CZA, CTZ, IMI/REL, MEV, and CFD were approved within a short period of time. All of these molecules offer new treatment options for multidrug-resistant bacteria and this impetus in research should continue with the imminent development of new β-lactamase inhibitors. Despite their promise, many uncertainties remain regarding their indications and therapeutic applications. Indeed, in the absence of high-quality studies on ecology, microbiology, clinical efficacy, and the emergence of resistance, they cannot be considered as carbapenem-sparing strategies and prescribed “larga manu” in a probabilistic context. Finally, cost-effectiveness analyses should address the relevance of these newly developed, expensive molecules.

## Figures and Tables

**Figure 1 antibiotics-12-00654-f001:**
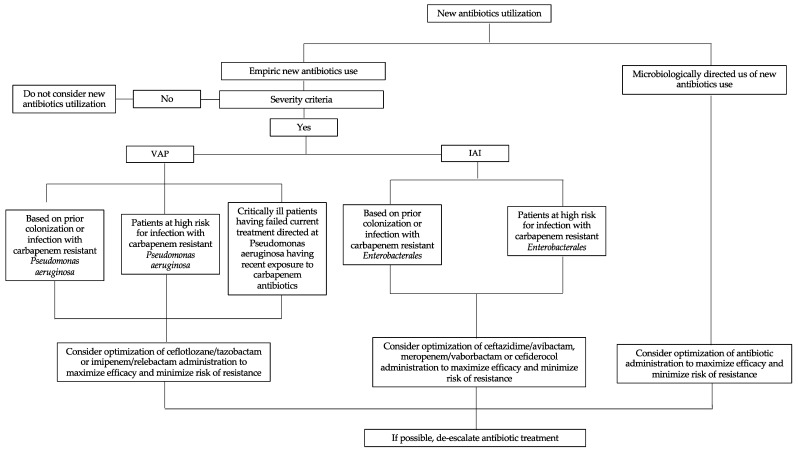
Proposed strategy for the clinical use of new antibiotics against Gram-negative bacteria. VAP: ventilator-associated pneumonia; IAI: intra-abdominal infection.

## Data Availability

Not applicable.

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
