# Peer review of "What to Do with the New Antibiotics?"

_antibiotics, 2023, doi:10.3390/antibiotics12040654_

Round 1

Reviewer 1 Report

The authors have done great efforts in compiling this review. This review will significantly contribute to address the AMR issues. The manuscript is written well and can be proceed further for publication process. I don’t have any major comments for this manuscript except some grammatical mistakes and unnecessary spaces in the main text between the words. My comments are:

1.     Line 35. Italicize “Enterobacteriaceae”.

2.     Line 288, 289, 292 etc: Make sure to italicize all of the organism and gene names throughout the manuscript.

3.     Line 59,127. Replace beta with β.

4.     Line 112,113,121: put a space between “P.aeruginosa

5.     Line 114-115: cite a reference here “Karlowsky et al.”

6.     Line 128-132: References are wrongly cited here. Remove extras and cite only one suitable.

7.     Line 147-152: References are wrongly cited here. Remove extras and cite only one suitable.

8.     Line 165: the authors have mentioned here about table 1, but there is no table 1 in the main manuscript. Please cross check.

9.     Line 218: References are wrongly cited.

10.  Line 252: This is not a good way to write the quotes here. Please remove. Authors can write about their idea and then cite.

11.  Line 303: Write the abbreviation after giving full stop to the previous sentence.

12.  Line 325: the authors have mentioned here about table 1, but there is no table 1 in the main manuscript. Please cross check.

13.  Line 338: Please make this figure as proper table or otherwise change this to figure by giving the colour coding.

14.  Line 357: Make the organism names italicize.

15.  Line 530: There should be np references in the conclusion section. The authors should give their own conclusion of this review.

16.  I suggest to add a new section of future recommendations.

17.  The authors needs to check the references again, and remove unnecessary citations.

Author Response

Reviewer 1

The authors have done great efforts in compiling this review. This review will significantly contribute to address the AMR issues. The manuscript is written well and can be proceed further for publication process. I don’t have any major comments for this manuscript except some grammatical mistakes and unnecessary spaces in the main text between the words. My comments are:

  1. Line 35. Italicize “Enterobacteriaceae”.

2.Line 288, 289, 292 etc: Make sure to italicize all of the organism and gene names throughout the manuscript.

  1. Line 59,127. Replace beta with β.
  2. Line 112,113,121: put a space between “P.aeruginosa
  3. Line 114-115: cite a reference here “Karlowsky et al.”
  4. Line 128-132: References are wrongly cited here. Remove extras and cite only one suitable.
  5. Line 147-152: References are wrongly cited here. Remove extras and cite only one suitable.
  6. Line 165: the authors have mentioned here about table 1, but there is no table 1 in the main manuscript. Please cross check.
  7. Line 218: References are wrongly cited.
  8. Line 252: This is not a good way to write the quotes here. Please remove. Authors can write about their idea and then cite.
  9. Line 303: Write the abbreviation after giving full stop to the previous sentence.
  10. Line 325: the authors have mentioned here about table 1, but there is no table 1 in the main manuscript. Please cross check.
  11. Line 338: Please make this figure as proper table or otherwise change this to figure by giving the colour coding.
  12. Line 357: Make the organism names italicize.
  13. Line 530: There should be np references in the conclusion section. The authors should give their own conclusion of this review.
  14. I suggest to add a new section of future recommendations.
  15. The authors needs to check the references again, and remove unnecessary citations.

We have made all the suggested changes

Reviewer 2 Report

The manuscript entitled as “What to do with the new antibiotics?” is an interesting and well-structured paper that will be of interest to the readers of this journal. The authors have tried to incorporate the different antibiotics with regard to epidemiological data. However, there are some minor corrections needed before the manuscript gets published in the journal Antibiotics:

1        The authors mentioned that “Multidrug resistant Gram-negative bacterial infections are a major global public health concern [1] leading to high morbidity and mortality [2,3]. Before discussing any specific bacterial strain, the authors should mention about both gram positive and gram negative bacteria or in general resistance.

2        There were grammatical mistakes were found. For example: line ‘Relebactam was found to significantly increases activity against imipenem non susceptible and β-lactamases (class A 111 enzymes) producing Enterobacterales and against P.aeruginosa (2- to 128-fold MIC reductions for Enterobacterales and eightfold MIC reduction for P.aeruginosa)’. The authors should check the whole manuscript for these mistakes.

3        For gram positive bacteria, cefiderocol has high MICs values for most known bacteria such as Staphylococcus aureus or Enterococcus spp [24]. In this line, authors have specifically mentioned about gram positive strain while the whole manuscript is centered towards the gram negative strain. Please explain.

Author Response

The manuscript entitled as “What to do with the new antibiotics?” is an interesting and well-structured paper that will be of interest to the readers of this journal. The authors have tried to incorporate the different antibiotics with regard to epidemiological data. However, there are some minor corrections needed before the manuscript gets published in the journal Antibiotics:

  • The authors mentioned that “Multidrug resistant Gram-negative bacterial infections are a major global public health concern [1] leading to high morbidity and mortality [2,3]. Before discussing any specific bacterial strain, the authors should mention about both gram positive and gram negative bacteria or in general resistance.

Thank you for this comment that allows us to improve our manuscript we have made the suggested changes.

  • There were grammatical mistakes were found. For example: line ‘Relebactam was found to significantly increases activity against imipenem non susceptible and β-lactamases (class A 111 enzymes) producing Enterobacterales and against P.aeruginosa (2- to 128-fold MIC reductions for Enterobacterales and eightfold MIC reduction for P.aeruginosa)’. The authors should check the whole manuscript for these mistakes.

Thank you for your comment we have check the whole manuscript for mistakes

  • For gram positive bacteria, cefiderocol has high MICs values for most known bacteria such as Staphylococcus aureus or Enterococcus spp [24]. In this line, authors have specifically mentioned about gram positive strain while the whole manuscript is centered towards the gram negative strain. Please explain.

We agree with your remark and have therefore removed this sentence

Reviewer 3 Report

In the review article titled “What to do with the new antibiotics” Khalil Chaibi et al has shown that drug resistance to many gram-negative bacteria are now a significant public health issue that highlight the possibility of a treatment dead end. Numerous new antibiotics have been developed recently, expanding the medicinal arsenal. The author of this review has concentrated on the position of various antibiotics in relation to epidemiological data. The study is of great interest and for its further improvement I have few minor suggestions/comments:

Minor Comments:

·      Line 61-62 In patients in the ICU patients, rewrite the sentence

·      Line 62 expand ICU

·      Line 75 expand ESBL

·      Line 96 expand AmpC

·      Line 98 0,5 Write as 0.5

·      Line 103-104 give reference

·      Line 112 write 2- as 2

·      Line 113 write eight in numerical 8

·      Line 148 expand GNB

·      Line 364-366 CTZ resistance requires………AmpC related genes rewrite the sentence for better clarity

·      Line 382-385 Resistance to CZA in…….efflux pumps rewrite the sentence for better clarity

Author Response

In the review article titled “What to do with the new antibiotics” Khalil Chaibi et al has shown that drug resistance to many gram-negative bacteria are now a significant public health issue that highlight the possibility of a treatment dead end. Numerous new antibiotics have been developed recently, expanding the medicinal arsenal. The author of this review has concentrated on the position of various antibiotics in relation to epidemiological data. The study is of great interest and for its further improvement I have few minor suggestions/comments:

Minor Comments:

  • Line 61-62 In patients in the ICU patients, rewrite the sentence

We changed as suggested

  • Line 62 expand ICU
  • Line 75 expand ESBL
  • Line 96 expand AmpC

We expand ICU, ESBL and AmpC as suggested

  • Line 98 0,5 Write as 0.5

We rewrote as suggested

  • Line 103-104 give reference

We changed as suggested

  • Line 112 write 2- as 2

We changed as suggested

  • Line 113 write eight in numerical 8

We changed as suggested

  • Line 148 expand GNB

We changed as suggested

  • Line 364-366 CTZ resistance requires………AmpC related genes rewrite the sentence for better clarity

We changed as suggested

  • Line 382-385 Resistance to CZA in…….efflux pumps rewrite the sentence for better clarity

We changed the sentence for better clarity

Reviewer 4 Report

First of all I would like to congratulate the authors for the paper "What to do with the new antibiotics", a comprehensive review regarding the newly discovered antibiotics and their indication of use.

My concerns regarding the article are minor and are related to the following:

1. Abbreviations. Please recheck the whole document regarding the use of abbreviations and their explanation on the first use in text. 

- line 62 - ICU

- line 74 - MIC50/90

- line 81 CTZ/TZ, I think there, you actually meant CTZ

-line 96 - ESBL

-line 176 - C/T

- line 488 - PK/PD

- KPC or KPC-Kp? 427

2. I would suggest some more explanations in the Introduction section regarding the classes of beta-lactamates (OXA 24, OXA 48, CTX-M, VIM, IMP, NDM, KPC, etc) where you can explain the classes, as in the rest of the article you talk and compare them.

Author Response

First of all I would like to congratulate the authors for the paper "What to do with the new antibiotics", a comprehensive review regarding the newly discovered antibiotics and their indication of use.

My concerns regarding the article are minor and are related to the following:

  1. Please recheck the whole document regarding the use of abbreviations and their explanation on the first use in text. 

Thank you for this comment that allow us to improve our manuscript. We check the whole document regarding the use of abbreviations

- line 62 – ICU

We changed as suggested

- line 74 - MIC50/90

We changed as suggested

- line 81 CTZ/TZ, I think there, you actually meant CTZ

We changed as suggested

-line 96 – ESBL

We changed as suggested

-line 176 - C/T

We changed as suggested

- line 488 - PK/PD

We changed as suggested

- KPC or KPC-Kp? 427

We changed as suggested

  1. I would suggest some more explanations in the Introduction section regarding the classes of beta-lactamates (OXA 24, OXA 48, CTX-M, VIM, IMP, NDM, KPC, etc) where you can explain the classes, as in the rest of the article you talk and compare them.

We add a sentence to clarify our manuscript